# Using Construction and Demolition Waste Materials to Develop Chip Seals for Pavements

Mohsen Shamsaei ⬤, Alan Carter *⬤ and Michel Vaillancourt

Department of Construction Engineering, École de Technologie Superieure, 1100 Notre-Dame Street West, Montréal, QC H3C 1K3, Canada; mohsen.shamsaei.1@ens.etsmtl.ca (M.S.); michel.vaillancourt@etsmtl.ca (M.V.)
* Correspondence: alan.carter@etsmtl.ca

**Abstract:** Construction and demolition waste (CDW) materials account for a considerable part of waste materials throughout the world. As these materials are not usually recycled, reusing them in construction projects is of major significance. In this study, recycled concrete, bricks, and glass were used as 100% aggregates of chip seal, which is a corrective or preventive pavement maintenance method. A cationic rapid setting (CRS-2) bitumen emulsion was also used to prepare the chip seal. Different tests, including the sand patch test, sweep test, British pendulum tester (BPT), interface bond, and Vialit test, were conducted. The results of these tests revealed that all these materials had sufficient aggregate embedment for vehicle speeds of more than 70 km/h, and the number of chips was less than 10%, indicating their good performance. All developed chip seals ranked as high skid resistance pavement at ambient temperature. The chip seals developed with concrete and glass showed the best adhesion with an asphalt pavement surface and an aggregate–bitumen adhesion at very cold and ambient temperatures due to the fact of their chemical compositions. Overall, using concrete aggregates to develop chip seals under different traffic loads is recommended. Finally, these findings can provide a novel approach for recycling CDW materials with low costs.

**Keywords:** chip seal performance; mechanical properties; pavement maintenance; construction and demolition waste; aggregate adhesion

## 1. Introduction

The quantity of construction and demolition waste (CDW) materials has been increasing due to the current fact of global urbanization [1]. The United States (US), the European Union (EU), and China have been ranked as among the large generators of CDW [2]. This CDW mostly consists of metals, glass, plastics, woods, asphalt shingles, and mineral waste from construction and demolition, such as concrete, tiles, and bricks [3]. The latest statistics, from 2018, show that the US, EU, and China generated 600, 372, and 1704 Mt, respectively [4]. The amounts of these waste materials are also important in Canada, where CDW accounts for almost 27% of municipal solid waste discharged into landfills [5]. In general, approximately 10 billion tons of CDW are generated all over the world annually, demanding a practical solution [6]. Although the US, EU, and Canada have higher recovery rates, China's rate is lower than 10% [7]. However, the recycling rates of mineral waste from construction are low for the US, EU, and Canada, and concrete and brick waste materials account for the highest proportion (approximately 59%) of CDW from building construction, and it is mostly discharged into landfills [3,8,9].

Apart from environmental pollution, this growing trend of CDW materials has caused serious issues for people all over the world. In the megacity of Shenzhen (China), approximately 10 Mt of CDW had accumulated in a landfill that collapsed, in 2015, causing a landslide that destroyed buildings and killed 73 people [10]. Another disaster occurred due to the presence of CDW in the city of Chennai (India) in 2015. The high quantity of these materials was identified as a reason for a flood that negatively affected more than 400 families' economic livelihoods owing to the subsequent infrastructural damages [11]. Moreover,

in Sri Lanka and Ethiopia, in 2017, landslides in CDW landfills killed 30 and 115 people, respectively [12]. Hence, low rates of CDW recycling can not only cause environmental pollution but pose dangers to people's lives.

As revealed in one study, a high quantity of CDW (almost 50%) can be recycled [9]; however, the type of CDW is of great importance. For instance, recycling most building materials using current methods can generate a considerable amount of greenhouse gases, including methane, N2O, CO2, and fluorinated gases, exacerbating air pollution [13]. Hence, reusing these materials, especially concrete, clay bricks, and tiles, in the construction industry, including road pavements, the base of sports grounds, noise buffer walls, and landscape construction [14,15], can not only be a practical way to recycle these waste materials, but it could reduce the need for using nonrenewable raw materials for construction projects [16,17].

Deteriorated pavement replacement is costly, and chip sealing and microsurfacing are two corrective or preventive pavement maintenance methods that can be applied fast and easily on existing pavements [18,19]. Chip sealing is one of the most cost-effective types of bituminous surface treatments that can be applied for both concrete and asphalt pavements [20]. A chip seal consists of a layer of bitumen emulsion (60–70% asphalt cement, deionized water, and emulsifying agent) and one or multiple layers of aggregates, which are compacted by rollers before setting the bitumen emulsion [21]. Therefore, using recycled aggregates in chip seals can be a practical solution for recycling a high quantity of CDW from building constructions, including waste concrete, bricks, and glass.

CDW materials have been used in different parts of pavement structures in recent studies. In one study, CDW aggregates, including mortar, ceramic, and clay waste materials, were used as fine and coarse aggregates to prepare concrete paving. The mechanical properties of the prepared concrete were evaluated, and the test results indicated that using these materials increased the apparent porosity and water absorption of concrete, resulting in compressive strength and splitting tensile strength reduction. However, using CDW materials up to 50% by weight of the aggregates could meet the minimum standard requirement for concrete paving [22]. CDW materials (up to 100% recycled concrete) were also used to prepare pervious concrete. Different properties of the developed concrete, including hydraulic conductivity, porosity, infiltration, hardened density, infiltration, compressive, flexural, and tensile strengths, elasticity modulus, Poisson's ratio, and abrasion resistance, were evaluated. The test results showed that increasing the percentages of recycled concrete increased the porosity and decreased the strength. However, this mixture could meet the minimum compressive strength requirement of Brazil [23]. Recycled glass (0–5 mm) was also used as a base course material in another study. The resilient modulus ($M_r$) and shear strength of the prepared base course were evaluated, and the results revealed that increasing the percentage of glass aggregates reduced the $M_r$ and shear strength of the base course. However, using up to 25% recycled glass (by weight) had negligible and marginal negative effects on the mechanical properties of the base course [24].

Reclaimed asphalt pavement (RAP), recycled rubber, and CDW materials were also utilized in hot mix asphalt (HMA) in some studies. One important property of HMA, which can be altered after using these waste materials, is the skid resistance. The BPT test results in different weather conditions revealed that using RAP as aggregates and crumb rubber as bitumen additive had sufficient skid resistance for road users, and in some weather conditions, the developed HMA showed better skid resistance in comparison to conventional HMA [25,26]. In addition, recycled glass was used as fine aggregates (up to 25% by weight) of HMA. The results of thermal cracking, stripping resistance, mixture stiffness, and rutting tests showed that using glass decreased the binder content, rutting resistance, and stripping resistance of HMA. However, it increased the mixture's workability, and using this material up to 10% had negligible effects on thermal cracking and mixture stiffness [27]. Mammeri et al. (2023) also replaced the virgin aggregates of HMA with recycled glass (100% by weight) to mitigate the UHI effects. Glass reduced the

surface temperature at night, and the control mixture absorbed and released 34% and 47% more heat in comparison to the HMA prepared with glass aggregates [28].

Chip seals have begun to gain researchers' attention in recent years, and a few studies have been conducted on the use of different types of aggregates for chip seals. In one study, trap-rock and granite were used to prepare chip seals, and their performance was evaluated with the Vialit test (cold temperatures) and Michigan Tech's interface bond test (IBT) for cold and normal temperatures. The results revealed that the durability of the chip seal is affected by the freeze–thaw cycle and asphalt–aggregate combinations. Hence, using suitable aggregate for chip seals in cold areas is of great importance [21]. Recycled crumb rubber was used as aggregates for the chip seal. The prepared chip seal was assessed with the sand patch method, skid test, and image processing technique. The results of these tests showed that crumb rubber had an adequate embedment depth and skid resistance, and natural aggregates can be partially or fully replaced with crumb rubber in chip seals [29]. Reclaimed asphalt pavement (RAP) was another recycled material used for chip seal preparation. The physical properties of RAP were evaluated based on chip seal standards, and the results indicated that RAP met the mix design requirements and can be used for chip sealing [30]. Therefore, despite the mentioned engineering and environmental benefits of using CDW materials for pavement construction, these materials have not been used for preparing chip seals yet.

This research is part of a comprehensive research program aiming to modify the thermal behavior of asphalt pavements to mitigate the urban heat island (UHI) effects. As some CDW materials have light colors, they may reduce the surface temperatures of asphalt pavements, alleviating the UHI effects [31]. Hence, the mechanical properties and the possibility of using some CDW materials for chip seal preparation were evaluated in this study. For this purpose, CDW materials, including concrete, clay bricks, and glass, were used as 100% of the chip seal aggregates. A cationic rapid setting (CRS-2) bitumen emulsion was also utilized for chip seal preparation. The goal of this study was to evaluate the effectiveness of those CDW materials as aggregate in chip seals with experimental methods. As these recycled aggregates have not been used for chip seals before, different required tests were conducted to evaluate the mechanical properties of the designed chip seals, including the sand patch test, sweep test, British pendulum tester (BPT), interface bond test, and Vialit test. These experiments revealed the performance of the developed chip seal in hot and cold weather conditions. According to previous studies, using these waste materials as chip seal aggregates has not been investigated yet. Hence, the results of this study can provide a new method for recycling a considerable quantity of CDW materials and enhancing chip seal performance. The different sections of this research are illustrated in Figure 1.

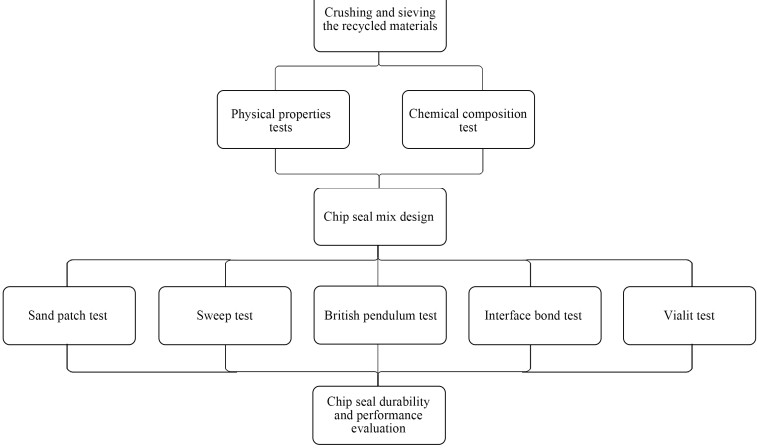

**Figure 1.** The research details.

## 2. Materials and Methods

### 2.1. Materials

A chip seal is composed of aggregates and bitumen emulsion. In this study, 4 different recycled aggregates, including concrete, yellow brick, red brick, and glass, were used. The waste materials were received from the Association of Building and Demolition Materials Collectors and Recyclers of Quebec (3RMCDQ) in Canada. The crushed materials are shown in Figure 2. The clay bricks (yellow and red) and concrete aggregates were in one range of coarse aggregates (Between 5 mm and 10 mm), and as all of them were crushed by the same device, their gradation was approximately similar. Compared to these aggregates, the glass aggregates were smaller (between 2.36 mm and 6.3 mm). Two reasons for this are attributed to the smaller sizes of the glass aggregates; the larger glass aggregates were flaky, and their larger sizes could damage the tires, so they were removed. The physical and chemical properties of these materials were tested based on relevant standards and the X-ray fluorescence (XRF) method, reported in Tables 1 and 2, respectively. In order to conduct the XRF tests, the recycled concrete aggregates, clay brick aggregates, and glass aggregates were crushed into powder (passing through Sieve 75 μm, No. 200). The results shown in Table 2 are the average of three repetitions, where the XRF test was performed on each sample with 3 repetitions. The obtained results are comparable to those found in the literature [32–34].

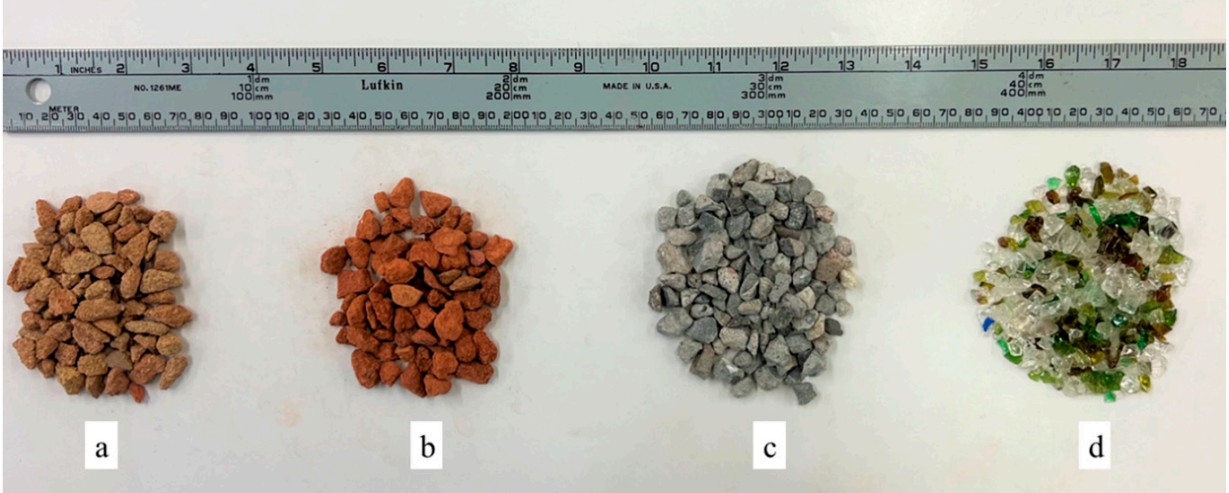

**Figure 2.** Crushed materials used in this study: (**a**) yellow clay brick; (**b**) red clay brick; (**c**) concrete; (**d**) glass.

**Table 1.** Physical properties of the recycled aggregates.

| Properties | Standard | Concrete | Yellow Brick | Red Brick | Glass |
|---|---|---|---|---|---|
| Density (kg/m$^3$) | ASTM C127 [35] | 2369 | 1932 | 1906 | 2490 |
| Loose unit weight (kg/m$^3$) | ASTM C29 [36] | 1513 | 1116 | 1121 | 1485 |
| Voids in loose aggregates (%) | ASTM C29 [36] | 35 | 58 | 59 | 36 |
| Water absorption (%) | ASTM C127 [35] | 4.7 | 13.55 | 14.12 | 0.15 |
| Los Angeles (%) | ASTM C131 [37] | 23 | 39 | 41 | 44 |
| Flakiness index (%) | ASTM D4791 [38] | 15 | 20 | 22 | 31 |
| Median particle size (mm) | ASTM C136 [39] | 6.5 | 6.7 | 6.8 | 4.7 |

**Table 2.** The chemical composition of the recycled aggregates.

| Chemical Composition (%) | Concrete | Yellow Brick | Red Brick | Glass |
|---|---|---|---|---|
| Silicon dioxide ($SiO_2$) | 34.97 | 59.03 | 62.07 | 73.30 |
| Aluminum oxide ($Al_2O_3$) | 7.34 | 14.53 | 16.14 | 1.68 |
| Iron (III) oxide ($Fe_2O_3$) | 3.31 | 6.44 | 5.07 | 0.38 |
| Calcium oxide (CaO) | 47.19 | 11.90 | 5.45 | 11.41 |
| Magnesium oxide (MgO) | 1.90 | 2.44 | 3.38 | 1.11 |
| Sulfur trioxide ($SO_3$) | 1.17 | 0.61 | 0.68 | 0.14 |
| Potassium oxide ($K_2O$) | 1.47 | 2.80 | 3.69 | 0.50 |
| Sodium oxide ($Na_2O$) | 1.34 | 1.10 | 2.34 | 10.70 |
| Titanium dioxide ($TiO_2$) | 0.40 | 0.72 | 0.75 | 0.04 |
| Phosphorus pentoxide ($P_2O_5$) | 0.13 | 0.12 | 0.16 | 0.02 |
| Loss on ignition (LOI) | 0.78 | 0.31 | 0.27 | 0.72 |

The properties of the cationic rapid setting type 2 (CRS-2) bitumen emulsion that was utilized for all of the chip seals are summarized in Table 3. This type of bitumen emulsion is reported as a common type for chip seal preparation. The recommended application temperature for this bitumen emulsion, as reported by the supplier, ranges from 60 to 85 °C. Hence, the temperature of the bitumen emulsion for all specimens in this study was 65 °C. The bitumen emulsion's water breakout was measured by its weight loss at 35 °C, as demonstrated in Figure 3. As can be seen, the bitumen emulsion lost approximately 25% of its weight after 6 h and 30% of its weight after 24 h due to the fact of water breakout.

**Table 3.** The bitumen emulsion properties.

| Tests | Unit | Test Method | Result For CRS-2 | Specifications Min. | Max. |
|---|---|---|---|---|---|
| Residue by distillation (by weight) | % | ASTM D6997 [40] | 69 | 65 | - |
| Oil distillate (by volume) | % | ASTM D6997 [40] | 0.3 | - | 3 |
| Demulsibility | % | ASTM D6936 [41] | 85 | 40 | - |
| Saybolt Furol viscosity at 50 °C | s | ASTM D7496 [42] | 223 | 100 | 400 |
| Settlement and storage stability, 1 day | % | ASTMD6930 [43] | 0.31 | | Max. 1 |
| Penetration, 25 °C, 100 g, 5 s. | 0/1 mm | ASTM D5 [44] | 153 | 100 | 250 |
| Particle charge | | ASTM D7402 [45] | Positive | | |
| Ductility, 4 °C, 5 cm/min | cms | ASTM D113 [46] | 75 | 40 | - |
| Solubility in TCE | % | ASTM D2042 [47] | 99.5 | 98.5 | - |

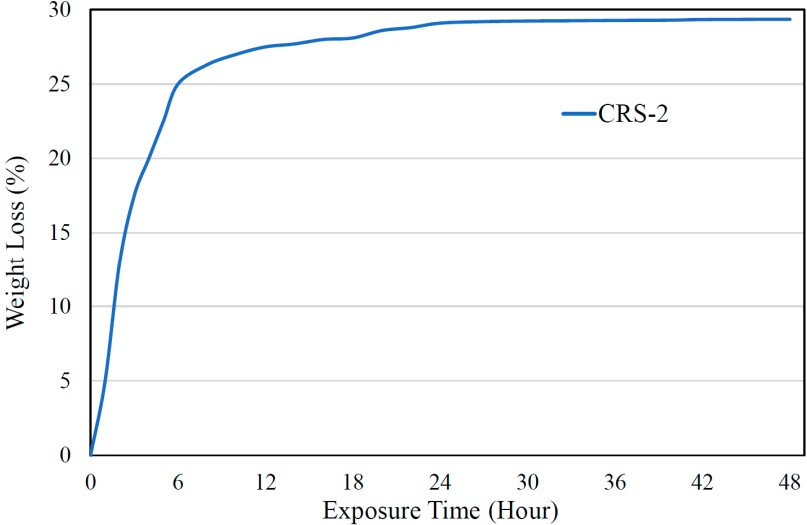

**Figure 3.** The bitumen emulsion's water breakout over the exposure time.

### 2.2. Chip Seal Design

There is no worldwide consensus on chip seal design. In most studies and real projects in the US, the McLeod, Kearby, modified Kearby, and modified McLeod methods are used to design chip seals [48–50]. The aim of these methods is to calculate the application rates of aggregates and bitumen emulsion to result in aggregate embedment ranging from 50% to 80% of the median aggregate size. The modified McLeod method, based on the Minnesota seal coat handbook, was used to design the chip seal in this study [51]. Based on the McLeod method, the aggregate rate depends on the aggregate shape, gradation, and specific gravity. Moreover, the binder rate depends on other factors, including the absorption and shape of aggregates, traffic volume, current pavement condition, and asphalt content of the binder. Hence, according to the experimental test on the recycled aggregates, bitumen emulsion, and McLeod's equations, the application rates of the concrete, yellow brick, red brick, and glass aggregates were 7.88 kg/m$^2$, 6.11 kg/m$^2$, 6.02 kg/m$^2$, and 4.85 kg/m$^2$, respectively. The emulsion rates for the concrete, yellow bricks, red bricks, and glass chip seals were 1.5 L/m$^2$, 1.88 L/m$^2$, 1.9 L/m$^2$, and 1.44 L/m$^2$. To check the calculated rates, some molds were prepared with the average least dimension (ALD) height, and the relevant aggregate and bitumen emulsion rates were used and compacted in these molds. These application rates filled the whole molds, and as the bitumen emulsion had approximately 30% water (Figure 3), the aggregate embedment was approximately 70% after the water breakout. However, the McLeod method assumes that a sufficient aggregate embedment depth will be provided after two years of service.

### 2.3. Description of Tests

#### 2.3.1. The Sand Patch Test

This test was used to evaluate the macrostructure of the chip seal. Considering the diameter of the specimens, which was 300 mm, the suitable amounts of aggregates and bitumen emulsion for each type of aggregate were calculated. According to the bitumen emulsion rates, as mentioned in Section 2.2, the amounts of bitumen emulsion for concrete, yellow brick, red brick, and glass aggregates were 109 g, 136 g, 137 g, and 104 g, respectively. Similarly, according to the area of the samples, suitable aggregate rates were used for each type of aggregate. Indeed, after applying the relevant amounts of bitumen emulsion on the asphalt felt disks, the concrete, yellow brick, red brick, and glass aggregates (557 g, 432 g, 425 g, and 343 g, respectively) were uniformly placed on the applied bitumen emulsion. The aggregates were then compacted by a standard compacter, which was 7500 g with a curved surface radius of 550 mm based on ASTM D7000 [52]. The prepared samples were then rotated (90°) to remove the loose aggregates. Two steps of curing were used to break out the emulsion's water. Firstly, the specimens were cured at 35 °C for 48 h, followed by 48 h of curing at ambient temperature. The sand patch test was conducted on the cured samples based on ASTM E965 to determine the mean texture depth (*MTD*) of twelve specimens (4 different aggregates with 3 repetitions) [53]. Based on this standard, a certain amount of Ottawa sand (75 mL in this study) should be spread uniformly in different orientations from the center of the sample with a stiff rubber object. The diameter of the spread sand was then measured. The *MTD* of the samples was calculated with Equation (1).

$$MTD = \frac{4V}{\pi D^2} \tag{1}$$

where *V* is the sand volume (mm$^3$), *D* is the spread diameter (mm), and *MTD* is the mean texture depth (mm). The specimen preparation steps and sand patch test method are depicted in Figure 4.

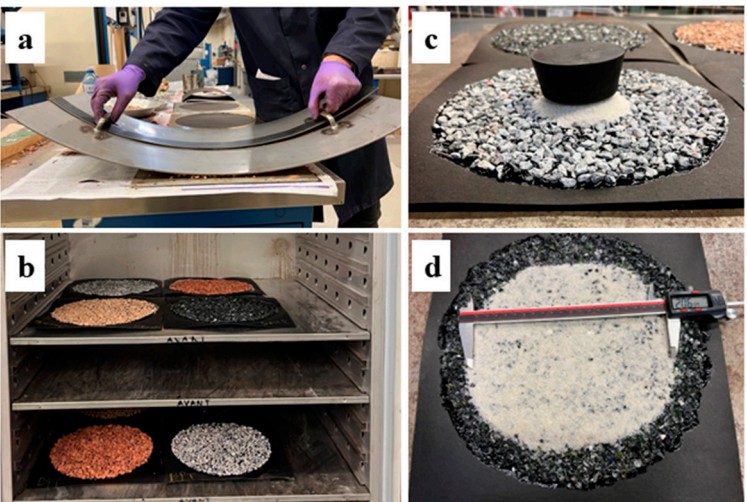

**Figure 4.** The sand patch test procedure: (**a**) sample preparation; (**b**) curing; (**c**) spreading the sand; (**d**) diameter measurement.

### 2.3.2. The Sweep Test

The chip seal performance in terms of aggregate loss was evaluated based on ASTM D7000 [52]. Asphalt felt disks with a diameter of 300 mm were cut and cured at 50 °C for 48 h. These disks were then cured at ambient temperate for 48 h to flatten them, and their weights were recorded. The sample preparation and amounts of materials were thus similar to the sand patch test, and twelve specimens were made. These specimens were then cured at 35 °C for 4 h. A brush was then used to remove the loose aggregates from the cured sample. Subsequently, the weights of these samples were recorded as the initial weights. Afterward, the sweep test apparatus, which was a modified mixer with a Nylon brush, was used at a rate of 0.83 gyrations per second for 60 s to measure the aggregate loss, which is demonstrated in Figure 5. In the next step, the loose aggregates were removed with a brush, and the specimens' weights were recorded as the final weights. Finally, the aggregate loss percentage was calculated with Equation (2).

$$Agreegate\ Loss\ (\%) = \left(\frac{A - B}{A - C}\right) \times 100 \times 1.33 \tag{2}$$

where *A* is the initial sample weight (g), *B* is the final sample weight (g), and *C* is the asphalt disk weight (g).

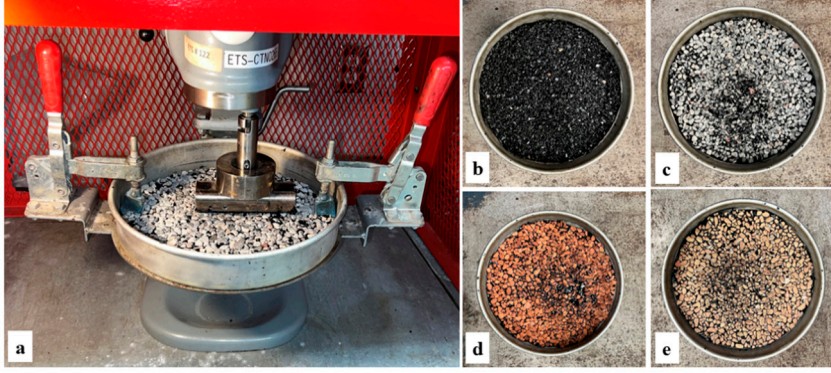

**Figure 5.** (**a**) The sweep test procedure: (**b**) glass; (**c**) concrete; (**d**) red brick; and (**e**) yellow brick specimens after the test.

### 2.3.3. The British Pendulum Tester

The friction value as a skid resistance parameter was measured with the British pendulum tester (BPT). According to the relevant standard for this test (ASTM E303), the specimen's dimensions should be at least 89 mm by 152 mm to provide sufficient contact area with the BRT pendulum. Hence, special steel molds were prepared, whose diameters were 100 mm by 170 mm to provide a flat surface and sufficient contact area. The appropriate amount of bitumen emulsion and aggregates were used to prepare the specimens based on chip seal designs. Hence, 26 g, 33 g, 33 g, and 25 g of bitumen emulsion were used for concrete, yellow brick, red brick, and glass, respectively. The aggregate rates for concrete, yellow brick, red brick, and glass were 134 g, 104 g, 102 g, and 82 g, respectively. After uniform aggregate distribution, a 5 kg roller compactor was used to compact the aggregates. Similar to the sand patch test, the specimens were firstly cured at 35 °C for 48 h, followed by 48 h of curing at the ambient temperature in order to break out bitumen emulsion water. As increasing the temperature raises the viscosity of bitumen, the chip seal aggregates may move under traffic loads in high temperatures, reducing the skid resistance of chip seals. Thus, considering the highest pavement temperatures in Canada [54], the skid resistance of the chip seals was evaluated at 60 °C and 23 °C with the BPT. Twenty-four specimens (3 repetitions for each type of aggregate) were made with different aggregates, and the British pendulum number (BPN) was recorded 5 times for each specimen, and the average BPN for each type of aggregate was calculated. For measuring the BPN at 60 °C, the specimen was placed on a heating plate for 2 h to reach 60 °C, after checking the chip seal temperature with a portable thermometer, the BPN values were measured and recorded. This test for the specimens at 23 °C and 60 °C is shown in Figure 6.

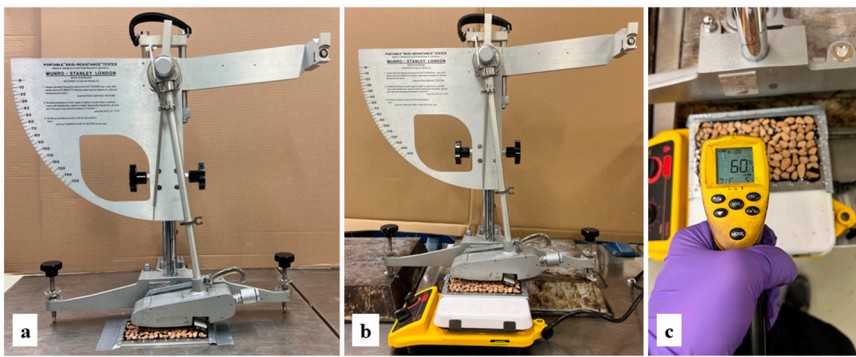

**Figure 6.** Skid resistance test: (**a**) at 23 °C; (**b**) at 60 °C; (**c**) chip seal temperature measurement.

### 2.3.4. The Interface Bond Test

The interface bond test for the chip seal was first developed at the Michigan Technological University [21]. This test was introduced as a suitable method for the investigation of chip seal and asphalt pavement interface bonds. The interface bond is of great importance, as premature stripping distress is attributed to a poor interface bond. Indeed, if the chip seal has a poor interface bond, it cannot be effective for pavement maintenance. In this test, thirty-six cylindrical asphalt concrete specimens (100 mm in diameter and 60 mm in height) were prepared, and chip seals containing recycled aggregates were applied on top of these specimens. Considering the area of the asphalt mixture samples, the relevant bitumen emulsion and aggregate rates were used. Hence, the aggregate rates for the concrete, yellow brick, red brick, and glass were 62 g, 48 g, 47 g, and 38 g, respectively. Similarly, the bitumen emulsion rates for these aggregates were 12 g, 15 g, 15 g, and 12 g, respectively. After pouring the bitumen emulsion on the asphalt mixture specimens and uniform aggregate distribution, a 5 kg roller compactor was used to compact the aggregates. Similar to the sand patch and BPT tests, the prepared specimens were first cured at 35 °C for 48 h, followed by 48 h of curing at an ambient temperature in order to break out the bitumen emulsion's water. Subsequently, a slow-setting epoxy adhesive, with 13 MPa

tensile strength, was applied at the top and bottom of the specimens to glue two aluminum caps. After 4 h of curing time for the epoxy, the specimens were placed at three different temperatures, −10 °C, 0 °C, and 23 °C, for 24 h to evaluate the interface bond strength in different weather conditions. Finally, a material testing system (MTS) device with a loading rate of 50 mm/min was used to measure the interface bond strength. Therefore, the peak load was divided into the cross-sectional area to calculate the interface bond strength of the chip seals with different aggregates. The test steps are demonstrated in Figure 7.

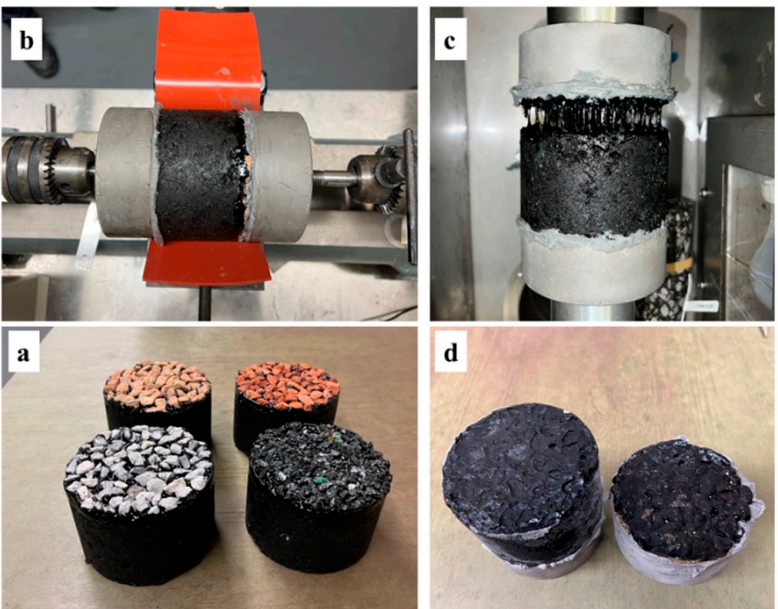

**Figure 7.** The interface bond strength test with the MTS device: (**a**) asphalt mixtures with chip seals; (**b**) gluing the caps on the specimen; (**c**) specimen during the test at 23 °C; (**d**) specimen after the test at −10 °C.

### 2.3.5. The Vialit Test

The adhesion of the bitumen emulsion and aggregates in cold climates was evaluated with the Vialit test. This test was conducted based on the BS EN 12272-3 standard [55]. As the adhesion between the aggregates and bitumen emulsion plays an important part in chip seal durability, this property was evaluated at −10 °C, −20 °C, and −30 °C for their durability during winter in Canada. Based on the BS standard, stainless-steel molds with diameters of 200 mm by 200 mm were used for this test. The relevant amounts of bitumen emulsion were spread on the molds, which were 62 g, 78 g, 78 g, and 59 g for the concrete, yellow brick, red brick, and glass aggregates. One hundred washed aggregates from each type of aggregate were placed on the distributed bitumen emulsion, followed by compaction with a 1 kg rubber roller. The samples were then cured at 35 °C for 48 h, followed by 48 curing at ambient temperature to break out the bitumen emulsion's water. Subsequently, the specimens were placed at −10 °C, −20 °C, and −30 °C for 4 h. In the next step, the molds were placed faced down and a steel ball (500 g) was dropped from a certain height (500 mm) three times on each mold. Thirty-six specimens were tested at these three temperatures, and the remaining aggregates on each mold were counted, the average of the values was used as the final retention ratio for each aggregate at each temperature. The Vialit test specimens, before and after the test, are illustrated in Figure 8.

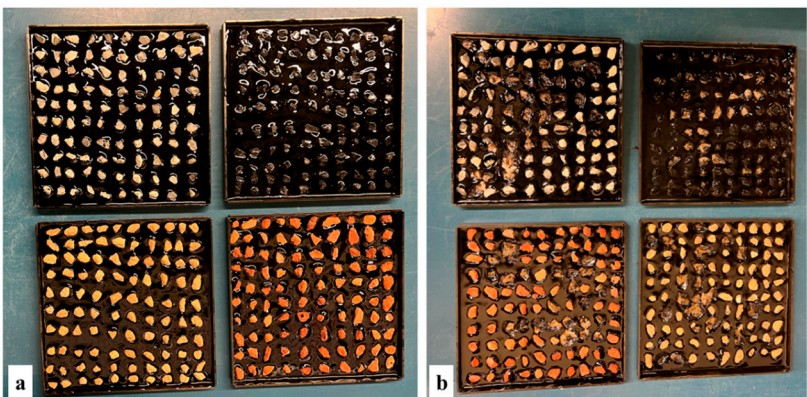

**Figure 8.** The Vialit test specimens: (**a**) before the test; (**b**) after the test.

## 3. Result and Discussion

### 3.1. The Macrotexture of Chip Seal

The MTD of the samples was determined with the volumetric sand patch method. The MTD values for four different types of aggregates are reported in Table 4. Based on the chip seal standards, the MTD should be more than 0.9 mm for posted speeds higher than 70 km/h and more than 0.7 mm for posted speeds lower than 70 km/h. If the MTD was lower than these values for the mentioned speeds, it was considered to be a chip seal failure [56]. Hence, the MTDs of all of the mix designs were more than the required limits. According to a US chip seal standard, depending on the aggregate sizes, the MTD of a typical chip seal is between 1 mm and 3 mm [57]. According to Table 4, the sand patch test revealed that the MTD values of all of these recycled materials were similar to the conventional materials. Unlike the concrete and brick aggregates, which have angular and irregular shapes, the recycled glass aggregates had partially flaky shapes for the used size (flakiness index = 31). Moreover, the used glass aggregates had smaller sizes in comparison to the other recycled materials. In addition, the glass aggregates had smoother surfaces compared to the concrete and brick aggregates. Hence, due to the abovementioned reasons, the MTD of the glass was lower than the other aggregates.

**Table 4.** The MTD values of different chip seals.

| Material | Avg. MTD (mm) | Variance | Standard Deviation (SD) | Coefficient of Variation (CV) (%) |
|---|---|---|---|---|
| Yellow brick | 2.70 | 0.0274 | 0.17 | 6.12% |
| Red brick | 2.68 | 0.0076 | 0.09 | 3.26% |
| Glass | 1.91 | 0.0107 | 0.10 | 5.42% |
| Concrete | 2.79 | 0.0336 | 0.18 | 6.58% |

### 3.2. The Sweep Test

The percentages of aggregate loss for the different materials are depicted in Figure 9. Based on a US standard for chip seals (NCHRP-Report 680), if the amount of chips dislodged is less than 10%, the chip seal has a suitable performance [58]. The aggregate loss was less than 10% for all different aggregates. The concrete aggregates had rough surfaces and angular and irregular shapes, resulting in sufficient adhesion between the concrete and bitumen emulsion, and only 7.43% of the aggerates were separated. The yellow and red brick aggregates had higher porosity in addition to rough surfaces and angular shapes, resulting in greater bitumen absorption and adequate aggregate–bitumen adhesion, showing 5.56%, and 7.05% aggregate loss. The glass aggregate had the least aggregate loss, only 2.36%. This may be because of the hydrogen bonds between the glass aggregates and bitumen emulsion due to the fact of its high contents of silica [21]. Moreover, the glass aggregates were smaller than the other aggregates and had a lower MTD, causing more

embedment depth. Moreover, lower air is trapped between the smaller aggregates and the asphalt pavement surface, raising the contact interface area, which can contribute to a lower aggregate loss [59].

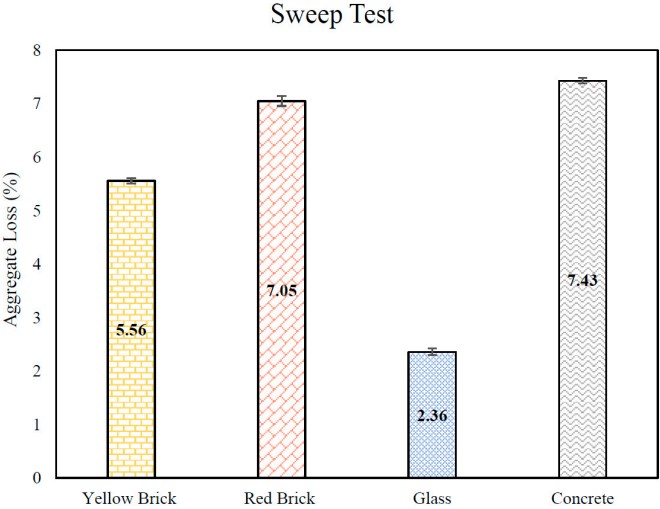

**Figure 9.** The percentages of aggregate loss for the different materials.

### 3.3. The Skid Resistance

The loss of skid resistance is common on pavements, and chip sealing is a practical way to increase the skid resistance [56]. Based on the standards and studies, the British pendulum number (BPN) over 55 represents a good skid resistance [60] and over 70 indicates a high skid resistance of the pavement [61]. The skid resistance depends on the surface macrostructure (texture wavelengths ranging from 0.5 mm to 50 mm and amplitude ranging from 0.1 mm to 20 mm) and microstructure (texture wavelength lower than 0.5 mm and amplitude lower than 0.2 mm) [62]. The chip seals' BPN values at 23 °C and 60 °C are reported in Table 5. The results reveal that all materials showed high skid resistance at 23 °C (BPNs over 70). The concrete aggregates also showed the highest skid resistance (BPN = 108), followed by yellow and red bricks, at 102 and 94, respectively. However, glass had the lowest BPN (74). These results are attributed to the MTD of the chip seals, as the concrete and bricks had higher MTD values in comparison to the glass. In addition, the glass aggregates had a higher flakiness index, resulting in lower surface roughness. Therefore, the BPN can indicate both the microstructure and macrostructure of the surface, which is in agreement with previous studies [60]. The correlation of the BPN and MTD at ambient temperature is shown in Figure 10. As can be seen, the $R^2$ value for these parameters is high (0.92). A comparison of the skid resistance at an ambient temperature (23 °C) and hot temperature (60 °C) is depicted in Figure 11. The skid resistance was reduced for all materials at 60 °C. The concrete aggregates showed the lowest reduction (22.3%), and the glass aggregates had the largest decrease (28.2%). Similarly, the yellow and red bricks' skid resistance declined by 26% and 23.4%, respectively. This decrease is attributed to the lower stiffness of the binder at 60 °C, changing the positions of the aggregates under the applied load and movement of the pendulum. However, the molds of the chip seals were heated for the BPT test, and in real situations, the aggregates are heated by solar energy, and the absorbed heat is transferred to the binder. As a result, the temperature of the binder is raised, causing a lower binder stiffness. The thermal conductivity of conventional concrete aggregate is similar to that of conventional materials [63], but bricks and glass have a lower thermal conductivity [64,65], which can reduce heat transfer and propagation to the binder. Hence, these insulation materials can be beneficial for hot regions' chip sealing.

**Table 5.** The skid resistance and BPN values for different materials.

| Temperature (°C) | Materials | BPN (Avg.) | Variance | Standard Deviation (SD) | Coefficient of Variation (CV) (%) |
|---|---|---|---|---|---|
| 23 | Yellow brick | 102 | 6.64 | 2.58 | 2.52 |
| | Red brick | 94 | 6.56 | 2.56 | 2.72 |
| | Glass | 74 | 5.36 | 2.32 | 3.14 |
| | Concrete | 108 | 6.64 | 2.58 | 2.39 |
| 60 | Yellow brick | 76 | 4.96 | 2.23 | 2.94 |
| | Red brick | 72 | 7.76 | 2.79 | 3.86 |
| | Glass | 53 | 6.80 | 2.61 | 4.92 |
| | Concrete | 84 | 3.44 | 1.85 | 2.22 |

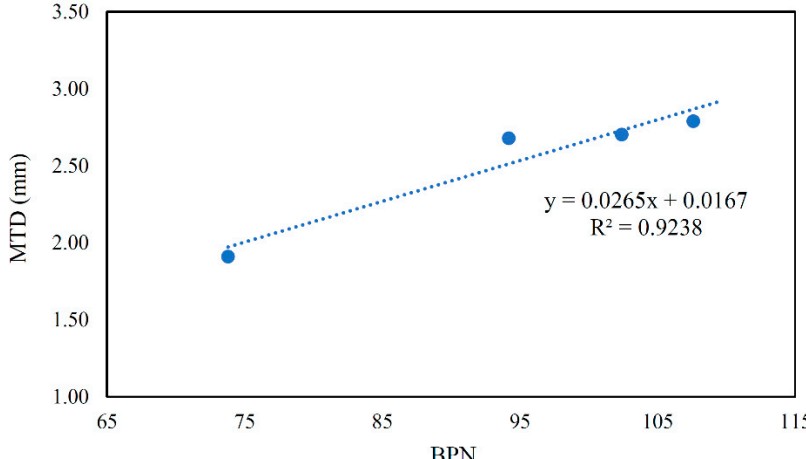

**Figure 10.** The correlation between BPN and MTD at the ambient temperature.

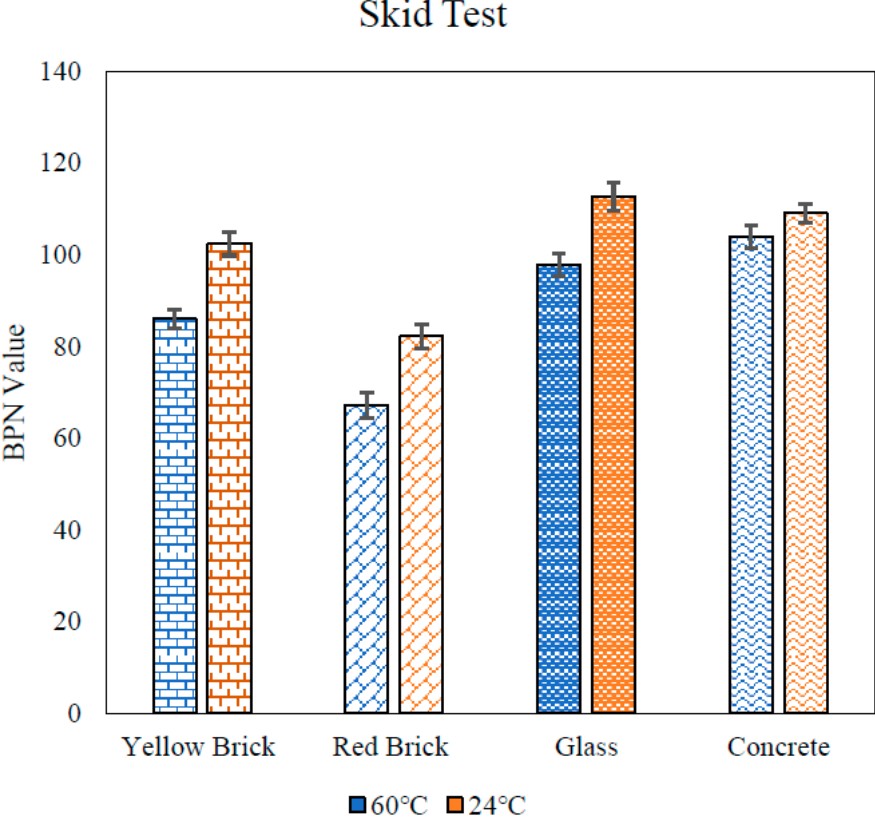

**Figure 11.** The comparison of BPN values at different temperatures.

*3.4. The Interface Bond between the Asphalt Mixture and Chip Seals*

The interface bond between chip seals and the asphalt pavement surface can indicate the durability of chip seals on the pavement. Indeed, a poor interface bond can cause asphalt pavement failure via premature stripping. Therefore, this parameter was tested in cold and ambient temperatures. The interface bond strengths of the chip seals with different materials are demonstrated in Figure 12. As can be seen, the chip seal made with recycled concrete had the highest interface bond strength at all temperatures in comparison to the other aggregates. This stronger bond between the chip seal was made with a concrete and asphalt mixture at all temperatures. The concrete aggregates showed the highest bond strength at 0 °C, approximately 1185 kPa. The interface bond decreased by 13% after the temperature was dropped to −10 °C. This reduction is attributed to the brittle behavior of bitumen at low temperatures. However, the lowest interface bond was observed at the ambient temperature (750 kPa). This is attributed to the softer bitumen at 23 °C, resulting in less strength in the direct tensile test. Compared to other aggregates, the higher interface bond of the concrete may be because of its higher calcium ion content on the concrete aggregate's surface (as reported in Table 2), developing more ionic bonds with bitumen's carbonyl groups. Regarding Table 2, concrete aggregates also have high silica contents, which can produce stronger hydrogen bonds with the bitumen emulsion. These results are also in agreement with previous studies [21,66,67]. Although brick aggregates had more angular shapes, the second-highest interface bond was related to glass aggregates. Similarly, the glass showed the highest strength at 0 °C, 1108 kPa, and it was reduced to 1012 kPa and 709 kPa at −10 °C and 23 °C, respectively. The interface bond strength of the glass was very close to that of the concrete, especially at −10 °C and 23 °C, and it was higher than for the bricks. There are two reasons attributed to these higher interface bonds. Firstly, the glass aggregates were smaller than the other aggregates, decreasing the air voids among the aggregates. In addition, this smaller size can reduce the air voids between the asphalt mixture surface and aggregate layer, increasing the interface contact between the glass chip seal and the asphalt mixture surface. The second reason could be the high percentages of silica in glass aggregates (Table 2), contributing to stronger hydrogen bonds between the aggregates and bitumen emulsion [21,66,67]. Although glass aggregates had smaller sizes than concrete aggregates, their smooth surface reduced the interface bond in comparison to the concrete aggregates.

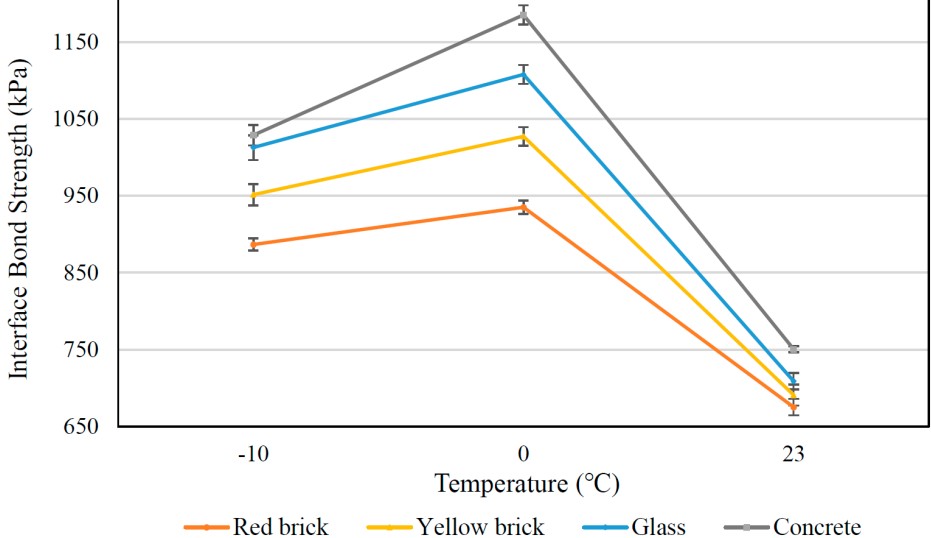

**Figure 12.** The interface bond for the chip seals at different temperatures.

The bricks had the lowest interface bond strength. Indeed, the yellow and red bricks' bond strengths were approximately 950 kPa at −10 °C and 0 °C, followed by an approxi-

mately 30% decrease at 23 °C due to the softer bitumen. Although the chemical composition and size of the aggregates play an important role in the interface bond strength of chip seals, unlike other materials, the adhesion between the brick aggregates and bitumen emulsion and asphalt surface was higher than the aggregate tensile strength. As a result, a few failures occurred in the brick aggregates during the interface bond strength test, which is demonstrated in Figure 13. Hence, the brick aggregates have good adhesion with the bitumen and asphalt pavement surface due to their rough and porous surface. Although the interface bond strengths of concrete and glass aggregates were much higher than bricks at −10 °C and 0 °C, the interface bond strengths of all materials were close to each other, at 23 °C, representing the importance of the bitumen stiffness of the chip seals in higher temperatures.

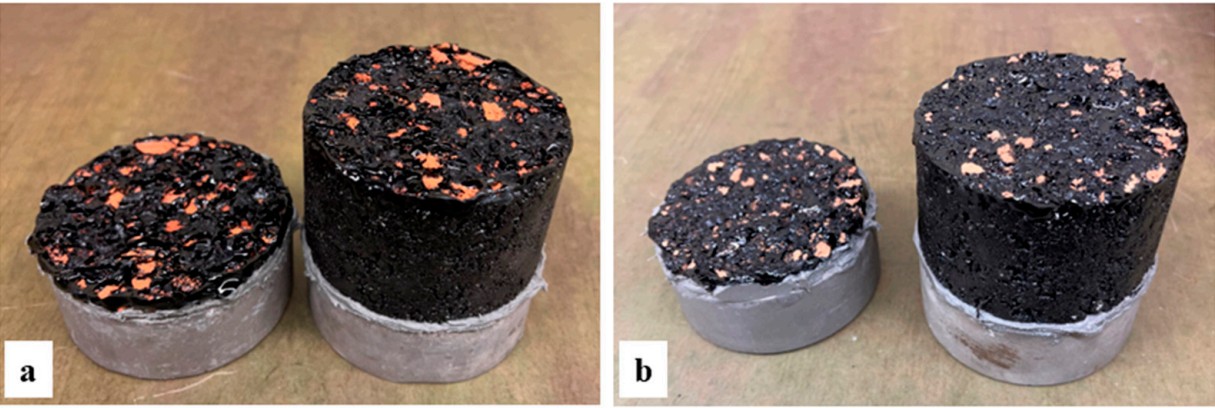

**Figure 13.** (**a**) Red and (**b**) yellow brick chip seals after the interface bond strength test.

*3.5. Aggregate and Bitumen Emulsion Adhesion in Low Temperatures*

The results of the Vialit test are depicted in Figure 14. These results revealed that dropping the temperature decreased the adhesion between the aggregates and bitumen emulsion for all aggregate types. This lower durability is attributed to the fragile behavior of bitumen after freezing. Hence, bitumen is brittle, and the aggregates are detached easier after the impacts. Regarding Figure 14, the glass and concrete aggregates had the highest retention ratio. The retention ratio of these two aggregates was approximately 75% at −10 °C. The glass retention ratio was reduced by 37% and 41% at −20 °C and −30 °C, respectively. The retention ratios of the concrete at −20 °C and −30 °C were reduced by 39% and 44%, respectively, which were very similar to the glass retention ratios. Regarding Table 2, this higher retention ratio in comparison to the brick aggregates is attributed to the higher calcium ion content of the concrete aggregates, producing stronger ionic bonds with the bitumen carbonyl group. In addition, the high silica contents of the glass may produce stronger hydrogen bonds between the aggregates and bitumen [21,66–68]. Another reason for the high retention ratio of the glass may be its smaller size in comparison to the other aggregates. As the interlayer area between the aggregates and bitumen can affect the bond strength, the impact resistance of smaller aggregates is higher than larger aggregates in the Vialit test. This result is in agreement with a previous study on the aggregate size impacts of chip seals [20]. The retention ratios of the yellow and red bricks were similar at −10 °C, approximately 69%. The retention values were almost 35% and 30% at −20 °C and −30 °C. In comparison to −10 °C, dropping the temperature from −20 °C to −30 °C had a lower negative effect on the retention ratios for all types of aggregates. The adhesion between the aggregates and bitumen of the chip seals at low temperatures is of major significance, as insufficient adhesion can cause premature stripping of the underneath asphalt pavements.

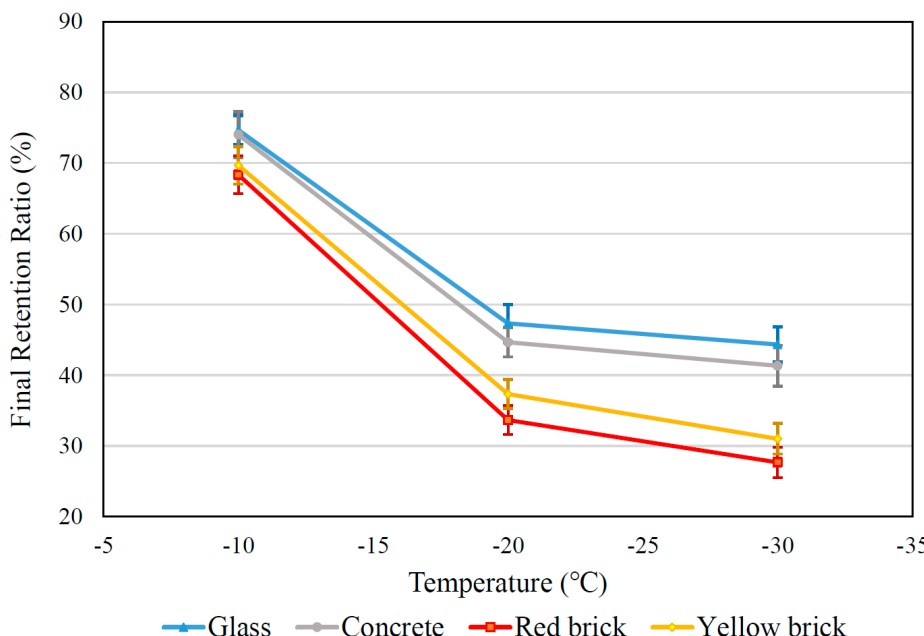

**Figure 14.** The retention ratios of different aggregates after the Vialit test.

One of the limitations of this study was related to the glass aggregate size. The aim was to use different recycled aggregates with similar sizes (between 5 and 10 mm). However, the glass aggregates within the mentioned size range had a high flaky index and could also damage the cars' tires. Hence, smaller sizes of glass aggregates were used. The other limitation can be related to the clay bricks and concrete aggregate sizes. Although the most common sizes of these CDW materials were used for chip seal preparation in this study, larger or smaller sizes may lead to different results. In addition, as the physical and chemical properties of the chip seal aggregates play significant roles in the chip seal's durability and performance, if other CDW materials (gypsum, wood, plastic, etc.) are used, different results may be achieved. Regarding the emulsion type, it is very important to use an emulsion with an electrical charge that is the opposite of the aggregates. If the same charge is used, the bitumen emulsion and aggregates will not bind well. As the charge of most aggregates is negative (Wood et al., 2006), it is important to use a cationic bitumen emulsion. Hence, the same results with similar cationic emulsions can be expected.

## 4. Conclusions

In this study, novel environmentally friendly chip seals were developed with CDW materials, including concrete, yellow and red clay bricks, and glass aggregates. The developed chip seals' properties were investigated to reveal essential information about the macrostructure, durability, skid resistance (in ambient and high temperatures), adhesion of the chip seals with the asphalt pavement at different temperatures, and the aggregate and bitumen emulsion adhesion in cold temperatures. The concrete and brick aggregates' sizes were also similar, and the glass aggregate size was smaller due to the fact of its flakiness index and the safety of cars' tires. Regarding the macrostructure of chip seals prepared with these CDW materials, the MTDs of all aggregates were in the standard range of chip seals, revealing that they had sufficient aggregate embedment to handle traffic speeds higher than 70 km/h. The laboratory sweep test also revealed that the chips dislodged were less than 10% for all aggregates, showing good chip seal performance with sufficient aggregate–bitumen adhesion. The glass aggregates had the lowest aggregate loss due to the fact of their chemical composition and smaller sizes. In addition, the BPN values of all of the aggregates were higher than 70, indicating a high skid resistance at the ambient temperature. Indeed, the concrete and glass aggregates had the highest and lowest skid resistance. Increasing the temperature to 60 °C decreased the skid resistance due to the fact

of the softer bitumen. However, all aggregates still had high skid resistance, except for the glass, which had an average skid resistance at high temperatures.

Moreover, the chip seals made with concrete aggregates and brick aggregates had the highest and lowest adhesion with the asphalt pavement at different temperatures, respectively. The chemical composition and size of the aggregates play an important role in the chip seal and asphalt pavement adhesion. Although both low and high temperatures can reduce the chip seal and asphalt pavement adhesion, a higher temperature has more negative effects on a chip seal's durability. Similarly, the aggregates' chemical compositions and sizes were reported as the most important parameters for the aggregate–bitumen adhesion of the chip seals. As the aggregate–bitumen adhesion was evaluated with a metal ball's impacts, the glass aggregates showed the highest aggregate–bitumen adhesion, followed by concrete aggregates with almost similar values.

Overall, all of the chip seals made with CDW materials showed a good performance in the different tests. However, as the abrasion resistance of the clay bricks and glass aggregates was not high, using these materials is recommended for low-volume roads, parking lots, driveways, and bicycle paths. In contrast, concrete aggregates not only have sufficient abrasion resistance and compressive strength, but the chip seal made with this aggregate also met all standard requirements and showed the highest values for almost all tests. Hence, using this aggregate for the chip sealing of different roads and paths is highly recommended. Finally, using all of these CDW materials as chip seal aggregates can provide a new way of recycling these waste materials, which are generated in huge quantities all over the world annually.

**Author Contributions:** Conceptualization, M.S., M.V. and A.C.; methodology, M.S.; software, M.S.; validation, M.S., M.V. and A.C.; formal analysis, M.S.; investigation, M.S.; resources, M.S.; data curation, M.S.; writing—original draft preparation, M.S.; writing—review and editing, M.V. and A.C.; visualization, M.S.; supervision, M.V. and A.C.; project administration, M.V. and A.C.; funding acquisition, M.V. and A.C. All authors have read and agreed to the published version of the manuscript.

**Funding:** The financial support of the Natural Sciences and Engineering Research Council of Canada (NSERC) is acknowledged (FRN: RGPIN-2020-04861).

**Data Availability Statement:** Data are available upon request to the corresponding author.

**Acknowledgments:** The authors would like to thank Ciment Quebec for performing the XRF tests on the recycled aggregates for us. We would also like to thank Sylvain Bibeau and Francis Bilodeau for their assistance and support with the laboratory work.

**Conflicts of Interest:** The authors declare no conflict of interest.

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
