# Peer review of "Using Construction and Demolition Waste Materials to Develop Chip Seals for Pavements"

_infrastructures, doi:10.3390/infrastructures8050095_

Round 1
Reviewer 1 Report
The reviewed manuscript "Using Construction and Demolition Waste Materials to Develop Chip Seals for Pavements", is an interesting study, raising current issues related to waste management in the construction sector. However, some elements of the paper should be improved:
1) The novelty and importance of this work should be emphasized in the abstract and the last paragraph of the Introduction section.
2) Numerical values in chemical compounds should be in indexes.
3) What is the demand for this work? Is this work helpful in practical applications?
4) The section related to the presentation of results and discussion of results should include more references to the literature (including the addition of recent references).
5) The methodology of the work needs more discussion.
6) The results section should be better defended using technical submitted and relevant references.
7) Conclusions should be refined and briefly presented. More numerical results should be added.
8) What are the limitations of the current research? Please list them in the manuscript.
Author Response
All authors sincerely wish to thank the reviewers for their comments, which have contributed significantly to the enhancement of the quality of the paper. Some information was added to the revised manuscript or revised based on your comments.
# Reviewer 1: The reviewed manuscript "Using Construction and Demolition Waste Materials to Develop Chip Seals for Pavements", is an interesting study, raising current issues related to waste management in the construction sector. However, some elements of the paper should be improved:
The novelty and importance of this work should be emphasized in the abstract and the last paragraph of the Introduction section.
Response: The following sentence was added to the abstract and the last paragraph of the Introduction section:
According to previous studies, using these waste materials as chip seal aggregates has not been investigated yet. Hence, the results of this study can propose a new method to recycle a considerable quantity of CDW materials and enhance the chip seal performance.
---------------------------------------------------------------------------------------------------------------------
2. Numerical values in chemical compounds should be in indexes.
Response: We agree that the information about the chemical composition may not be critical to this paper, but we prefer to leave it in the core of the paper since this information is used in the interface bond test analysis.
---------------------------------------------------------------------------------------------------------------------
3. What is the demand for this work? Is this work helpful in practical applications?
Response: As it is mentioned in the last paragraph of the Introduction section, this research is part of a comprehensive research program, aiming to modify the thermal behavior of asphalt pavements to mitigate the urban heat island (UHI) effects. The most important feature of these materials is their lighter colors which can increase the asphalt pavement surface reflectivity. Besides, using these easily available CDW materials for which there is a very limited demand can propose a new method for recycling these materials whose generation has been increasing in recent years. Hence, this study evaluates the possibility of using these materials as chip seal aggregates in terms of mechanical properties, safety, and chip seal overall performance. Considering the results of this study and depending on the traffic volume of the road, these materials can be used in real chip seal construction projects.
---------------------------------------------------------------------------------------------------------------------
4. The section related to the presentation of results and discussion of results should include more references to the literature (including the addition of recent references).
Response: Some relevant information and references were added to verify the results as follows:
Moreover, lower air is trapped between the smaller aggregates and the asphalt pavement surface, raising the contact interface area. which can contribute to a lower aggregate loss. This means that the aggregates’ sizes play an important role in aggregate retention in cold temperatures
You, L.; Jin, D.; You, Z.; Dai, Q.; Xie, X.; Washko, S.; Cepeda, S. Laboratory Shear Bond Test for Chip-Seal under Varying Environmental and Material Conditions. International Journal of Pavement Engineering 2021, 22, 1107–1115, doi:10.1080/10298436.2019.1662903.
Cui, P.; Wu, S.; Xiao, Y.; Hu, R.; Yang, T. Environmental Performance and Functional Analysis of Chip Seals with Recycled Basic Oxygen Furnace Slag as Aggregate. Journal of Hazardous Materials 2021, 405, 124441, doi:10.1016/j.jhazmat.2020.124441
---------------------------------------------------------------------------------------------------------------------
- The methodology of the work needs more discussion.
Response: It was revised, and a new flowchart was added to clarify the research methodology.
---------------------------------------------------------------------------------------------------------------------
- The results section should be better defended using technical submitted and relevant references.
Response: Some relevant information and references were added to verify the results as follows:
You, L.; Jin, D.; You, Z.; Dai, Q.; Xie, X.; Washko, S.; Cepeda, S. Laboratory Shear Bond Test for Chip-Seal under Varying Environmental and Material Conditions. International Journal of Pavement Engineering 2021, 22, 1107–1115, doi:10.1080/10298436.2019.1662903.
Cui, P.; Wu, S.; Xiao, Y.; Hu, R.; Yang, T. Environmental Performance and Functional Analysis of Chip Seals with Recycled Basic Oxygen Furnace Slag as Aggregate. Journal of Hazardous Materials 2021, 405, 124441, doi:10.1016/j.jhazmat.2020.124441
---------------------------------------------------------------------------------------------------------------------
- Conclusions should be refined and briefly presented. More numerical results should be added.
Response: It was implemented.
---------------------------------------------------------------------------------------------------------------------
- What are the limitations of the current research? Please list them in the manuscript.
Response: One of the limitations of this study was related to the glass aggregate size. The aim was to use different recycled aggregates with similar sizes (between 5 and 10 mm). However, the glass aggregates within the mentioned size range had a high flaky index and could also damage the cars’ tires. Hence, the smaller sizes of glass aggregates were used, which is explained in Section 2.1. The other limitation can be related to the clay bricks and concrete aggregate sizes. Although the most common sizes of these CDW materials were used for chip seal preparation in this study, larger or smaller sizes may bring about different results. Besides, as the physical and chemical properties of chip seal aggregates play significant roles in chip seal durability and performance if other CDW materials (such as gypsum, wood, plastic, etc..) are used, different results may be achieved. Regarding the emulsion type, it is very important to use an emulsion whose electrical charge is the opposite of aggregates. If the same charge of them is used, the bitumen emulsion and aggregates will not bind well. As the charge of most aggregates is negative (Wood et al., 2006), it is important to use cationic bitumen emulsion. Hence, the same results with similar cationic emulsions are expected. These limitations were added to the discussion section.
---------------------------------------------------------------------------------------------------------------------
Reviewer 2 Report
The following comments are to be considered while preparing the revised manuscript:
1. Abstract: For a journal article, the size of the abstract is to be presented within 150-200 words. Hence, the abstract is to be revised within the prescribed word limit by removing redundant/generic information.
2. Please clarify the rationale behind the recommendation of using concrete aggregates for developing chip seals
3. What is the novelty of the proposed work in the context of application perspective? Modification using a mere material change cannot be a major contribution in the proposed area? Please clarify.
4. Most of the information given in Table 1 are wrong. It is surprising that the authors have not known what is the specific gravity of concrete and other materials? Moreover, S.G does not have any units. Also, it is very well known that the unit weight of concrete is 2400 kg/m3. Moreover, the title of the table describes "Properties of aggregate" whereas the inside portion specifies different materials. Check all the data used in Table 1 and provide justification.
5. Again, Table 2 is also incorrect and requires adequate justification.
Author Response
All authors sincerely wish to thank the reviewers for their comments, which have contributed significantly to the enhancement of the quality of the paper. Some information was added to the revised manuscript or revised based on your comments.
Reviewer #2:
The following comments are to be considered while preparing the revised manuscript:
- Abstract: For a journal article, the size of the abstract is to be presented within 150-200 words. Hence, the abstract is to be revised within the prescribed word limit by removing redundant/generic information.
Response: The abstract was summarized in the revised version (200 words).
---------------------------------------------------------------------------------------------------------------------
- Please clarify the rationale behind the recommendation of using concrete aggregates for developing chip seals.
Response: As mentioned in the conclusion, concrete aggregates have very good results for most tests. This, by itself, is enough to recommend their use. It is mentioned in the last paragraph of the Introduction section that this research is part of a comprehensive research program, aiming to modify the thermal behavior of asphalt pavements to mitigate the urban heat island (UHI) effects. The most important feature of these materials is their lighter colors which can increase the asphalt pavement surface reflectivity. Besides, using concrete waste aggregates can propose a new method for recycling these materials whose generation has been increasing in recent years due to urbanization and population growth all over the world.
---------------------------------------------------------------------------------------------------------------------
- What is the novelty of the proposed work in the context of application perspective? Modification using a mere material change cannot be a major contribution in the proposed area? Please clarify.
Response: There are limited publications on chip seals, and even less on chip seals design with recycled materials. Because of that, we believe that the use of CDW as aggregates in chip seals is an important novelty here. Also, the link between some of the mechanical results with the chemical composition of the recycled aggregates is also a novelty.
---------------------------------------------------------------------------------------------------------------------
- Most of the information given in Table 1 are wrong. It is surprising that the authors have not known what is the specific gravity of concrete and other materials? Moreover, S.G does not have any units. Also, it is very well known that the unit weight of concrete is 2400 kg/m3. Moreover, the title of the table describes "Properties of aggregate" whereas the inside portion specifies different materials. Check all the data used in Table 1 and provide justification.
Response: The Table was modified. There was a translation error for specific gravity since it was meant to be Density. Also, the title of the Table was modified to have “recycled aggregates” properties. As for the values in Table 1, they are similar to these references (Lu et al., 2019; Gonzalez-Corominas and Etxeberria, 2016; Mammeri et al., 2022; Chen et al., 2019; Tareq Noaman et al., 2021).
Alducin-Ochoa, J.M., Martín-del-Río, J.J., Torres-González, M., Flores-Alés, V., Hernández-Cruz, D., 2021. Performance of mortars based on recycled glass as aggregate by accelerated decay tests (ADT). Construction and Building Materials 300, 124057. https://doi.org/10.1016/j.conbuildmat.2021.124057
Buss, A.F., Guriguis, M., Claypool, B., Gransberg, D.D., Williams, R.C., Oregon. Dept. of Transportation. Research Section, 2016. Chip seal design and specifications : final report. (No. FHWA-OR-RD-17-03).
Chen, F., Wu, K., Ren, L., Xu, J., Zheng, H., 2019. Internal Curing Effect and Compressive Strength Calculation of Recycled Clay Brick Aggregate Concrete. Materials 12, 1815. https://doi.org/10.3390/ma12111815
Gonzalez-Corominas, A., Etxeberria, M., 2016. Effects of using recycled concrete aggregates on the shrinkage of high performance concrete. Construction and Building Materials 115, 32–41. https://doi.org/10.1016/j.conbuildmat.2016.04.031
Lu, J.-X., Yan, X., He, P., Poon, C.S., 2019. Sustainable design of pervious concrete using waste glass and recycled concrete aggregate. Journal of Cleaner Production 234, 1102–1112. https://doi.org/10.1016/j.jclepro.2019.06.260
Mammeri, A., Vaillancourt, M., Shamsaei, M., 2022. Experimental and Numerical Investigation of Using Waste Glass Aggregates in Asphalt Pavement to Mitigate Urban Heat Islands (preprint). In Review. https://doi.org/10.21203/rs.3.rs-1862123/v1
Tareq Noaman, A., Subhi Jameel, G., Ahmed, S.K., 2021. Producing of workable structural lightweight concrete by partial replacement of aggregate with yellow and/or red crushed clay brick (CCB) aggregate. Journal of King Saud University - Engineering Sciences 33, 240–247. https://doi.org/10.1016/j.jksues.2020.04.013
Wood, T.J., Janisch, D.W., Gaillard, F.S., 2006. Minnesota seal coat handbook 2006.
Ye, T., Xiao, J., Zhao, W., Duan, Z., Xu, Y., 2022. Combined use of recycled concrete aggregate and glass cullet in mortar: Strength, alkali expansion and chemical compositions. Journal of Building Engineering 55, 104721. https://doi.org/10.1016/j.jobe.2022.104721
---------------------------------------------------------------------------------------------------------------------
- Again, Table 2 is also incorrect and requires adequate justification.
Response: Same as Table 1, the title of Table 2 was modified to add the mention recycled aggregates. Table 2 is obtained from the X-ray fluorescence (XRF) test. The recycled concrete aggregates, clay brick aggregates, and glass aggregates were turned to powder (passing from Sieve 75µm, No. 200). The prepared powders were sent to Ciment Quebec (https://www.cimentquebec.com/) where the XRF test was done on each sample with 3 repetitions. This factory has one of the most equipped laboratories in Canada and the XRF device is regularly calibrated and tested for cement XRF analysis. Besides, the XRF analysis of recycled concrete aggregates, bricks aggregates, and glass aggregates is compared with previous studies mentioned below, and the results are similar to our results. However, a little difference is normal due to the type of materials used to prepare these materials all over the world (Alducin-Ochoa et al., 2021; Tareq Noaman et al., 2021; Ye et al., 2022). This information was added to Section 2.1 to clarify this Table.
Alducin-Ochoa, J.M., Martín-del-Río, J.J., Torres-González, M., Flores-Alés, V., Hernández-Cruz, D., 2021. Performance of mortars based on recycled glass as aggregate by accelerated decay tests (ADT). Construction and Building Materials 300, 124057. https://doi.org/10.1016/j.conbuildmat.2021.124057.
Tareq Noaman, A., Subhi Jameel, G., Ahmed, S.K., 2021. Producing of workable structural lightweight concrete by partial replacement of aggregate with yellow and/or red crushed clay brick (CCB) aggregate. Journal of King Saud University - Engineering Sciences 33, 240–247. https://doi.org/10.1016/j.jksues.2020.04.013.
Ye, T., Xiao, J., Zhao, W., Duan, Z., Xu, Y., 2022. Combined use of recycled concrete aggregate and glass cullet in mortar: Strength, alkali expansion and chemical compositions. Journal of Building Engineering 55, 104721. https://doi.org/10.1016/j.jobe.2022.104721.
Reviewer 3 Report
The reviewer found the content of the paper very important and useful for minor rehabilitation measures that usually take place on the very top of pavements in favor of road safety. Some issues have been detected that require the authors’ attention:
1_ Lines 20-21 vs 100-101: Please correct the abstract, texture and skid resistance are not considered as mechanical properties. In lines 100-101, the authors are correctly mentioning some of the mechanical properties. Please be consistent in some terminology issues.
2_ Given the focus of the paper (i.e., chip seals), the reviewer believes that lines 115-127 are out of the scope and should be drastically limited. In the meantime, lines 99-114 should be enriched with more findings from the serviceability condition of wearing courses and surfacings (i.e, texture, skid, etc. – you may see and cite https://doi.org/10.3390/recycling7040047 , https://doi.org/10.3390/vehicles2010004 , or other similar ones.). In its current form, in lines 99-114 the focus is being put only on mechanical properties and relevant findings that could be kept of course, but in a more limited extent. Even acknowledging that findings from the functional performance of mixtures with recycling materials and CDW are sparse and limited can raise the motivation and the research value of the study.
3_ At the beginning of the methodology in section 2, please add a flowchart to illustrate the research outline. For example, type of testing, replicate specimens, etc.
4_ Line 199: I would add even in a worldwide scale, there are margins for improvements in the design of chip seals.
5_ Figure 10: Please correct the title in the vertical axis.
6_ Table 4 and Figure 8: The performance of different CDW is presented. Glass exhibits a lower performance. In the international literature, the use of glass particles has been also commented as somehow risky in terms of safety and injuries because of the aggregate dislodge. So, is it finally recommended? Please comment with research insights on this issue from the current investigation and the analysis results.
7_ It is not clear for the reviewer what conditions were followed to ensure temperature homogeneity during the measurements of skid resistance (preconditioning, holes in the slabs, etc). Please elaborate. Also, for what climatic conditions did the selected temperatures correspond to?
8_ Lines 548-549: It is probably too late to include statements with citing articles in the conclusions. Please move it earlier, perhaps in the introduction.
Overall, the paper needs to be revised with more efforts in the introduction (i.e., make the content more relevant) and a more broadened discussion of the results.
English is fine in general. Please make some cross-check for minor typos and language audit.
Author Response
All authors sincerely wish to thank the reviewers for their comments, which have contributed significantly to the enhancement of the quality of the paper. Some information was added to the revised manuscript or revised based on your comments.
Reviewer #3:
The reviewer found the content of the paper very important and useful for minor rehabilitation measures that usually take place on the very top of pavements in favor of road safety. Some issues have been detected that require the authors’ attention:
- Lines 20-21 vs 100-101: Please correct the abstract, texture and skid resistance are not considered as mechanical properties. In lines 100-101, the authors are correctly mentioning some of the mechanical properties. Please be consistent in some terminology issues.
Response: It was revised and removed from the abstract.
---------------------------------------------------------------------------------------------------------------------
- Given the focus of the paper (i.e., chip seals), the reviewer believes that lines 115-127 are out of the scope and should be drastically limited. In the meantime, lines 99-114 should be enriched with more findings from the serviceability condition of wearing courses and surfacings (i.e, texture, skid, etc. – you may see and cite https://doi.org/10.3390/recycling7040047 , https://doi.org/10.3390/vehicles2010004 , or other similar ones.). In its current form, in lines 99-114 the focus is being put only on mechanical properties and relevant findings that could be kept of course, but in a more limited extent. Even acknowledging that findings from the functional performance of mixtures with recycling materials and CDW are sparse and limited can raise the motivation and the research value of the study.
Response: The mentioned paragraph (lines 115-127) was removed. The other paragraph (99-114) was summarized, and the relevant references were used to enrich the information about recycled pavement materials.
---------------------------------------------------------------------------------------------------------------------
- At the beginning of the methodology in section 2, please add a flowchart to illustrate the research outline. For example, type of testing, replicate specimens, etc.
Response: It was added to the manuscript.
---------------------------------------------------------------------------------------------------------------------
- Line 199: I would add even in a worldwide scale, there are margins for improvements in the design of chip seals.
Response: We agree, the sentence was corrected.
---------------------------------------------------------------------------------------------------------------------
- Figure 10: Please correct the title in the vertical axis.
Response: It was revised.
---------------------------------------------------------------------------------------------------------------------
- Table 4 and Figure 8: The performance of different CDW is presented. Glass exhibits a lower performance. In the international literature, the use of glass particles has been also commented as somehow risky in terms of safety and injuries because of the aggregate dislodge. So, is it finally recommended? Please comment with research insights on this issue from the current investigation and the analysis results.
Response: Table 4 represents the MTD of chip seals developed with recycled aggregates. As it is mentioned in Section 3.1, according to a US chip seal standard, depending on the aggregate sizes, the MTD of a typical chip seal is between 1 mm and 3 mm (Buss et al., 2016). The glass’s MTD (1.91) is in the standard range and this lower value is attributed to its smaller size and smoother surface in comparison to other aggregates. Considering Figure 8 (it is Figure 9 in the revised version), glass had the lowest aggregate loss in the sweep test, which is desirable to have better performance. This study is also one part of a comprehensive study for changing the thermal behavior of asphalt pavement to mitigate the urban heat island. Hence, other aspects of glass aggregates will be published in another study. Besides, glass has a low thermal conductivity and can insulate the surface of the chip seal. As a result, less heat is transferred to the bitumen which will remain stiffer in hot seasons. The only problem with glass is its lower abrasion resistance. Thus, it is mentioned in the last paragraph of the conclusion, it is recommended to use glass for low-volume roads, parking lots, driveways, and bicycle paths.
---------------------------------------------------------------------------------------------------------------------
- It is not clear for the reviewer what conditions were followed to ensure temperature homogeneity during the measurements of skid resistance (preconditioning, holes in the slabs, etc). Please elaborate. Also, for what climatic conditions did the selected temperatures correspond to?
Response: The skid resistance was measured at two different temperatures, room temperature (23℃) and 60 ℃. The relevant samples were placed in the room for 2 hours at 23 ℃ temperature prior to the test. As it is explained in Section 2.3.3 and shown in Figure 6, the samples were placed on the heating plate with a control temperature gauge for 2 hours before the test. The surface temperatures of the samples were then measured with a portable thermometer (5 different spots for each sample). As chip seals are comprised of one layer of bitumen and one layer of aggregates placed inside the bitumen, using thermocouples was not possible for skid resistance samples. Regarding the selected temperatures, room temperature is the common temperature to measure the skid resistance in the laboratory. Also, as the bitumen loses its stiffness at high temperatures, resulting in aggregate dislodging of chip seals, 60 ℃ was chosen based on the highest pavement temperatures recorded in Canada to simulate the worst temperature condition for chip seals.
---------------------------------------------------------------------------------------------------------------------
- Lines 548-549: It is probably too late to include statements with citing articles in the conclusions. Please move it earlier, perhaps in the introduction.
Response: It was revised. It was moved to the last paragraph of the Introduction.
---------------------------------------------------------------------------------------------------------------------
- Overall, the paper needs to be revised with more efforts in the introduction (i.e., make the content more relevant) and a more broadened discussion of the results.
Response: Thank you for your comment. After revising the paper based on your comments and other reviewers’ comments, these issues were solved.
---------------------------------------------------------------------------------------------------------------------
Round 2
Reviewer 2 Report
Comments were addressed.
Reviewer 3 Report
The paper was improved.
It is fine.